# 3D Locating System for Pests' Laser Control Based on Multi-Constraint Stereo Matching

Yajun Li [1,2], Qingchun Feng [2,*], Jiewen Lin [3], Zhengfang Hu [1], Xiangming Lei [1] and Yang Xiang [1,*]

[1] College of Mechanical and Electrical Engineering, Hunan Agriculture University, Changsha 410128, China; lyj20210043@stu.hunau.edu.cn (Y.L.); victoria@stu.hunau.edu.cn (Z.H.); lxm@stu.hunau.edu.cn (X.L.)

[2] Intelligent Equipment Research Center, Beijing Academy of Agriculture and Forestry Sciences, Beijing 100097, China

[3] College of Engineering, China Agricultural University, Beijing 100083, China; b20203070536@cau.edu.cn

* Correspondence: fengqc@nercita.org.cn (Q.F.); xy@hunau.edu.cn (Y.X.)

**Abstract:** To achieve pest elimination on leaves with laser power, it is essential to locate the laser strike point on the pest accurately. In this paper, *Pieris rapae* (L.) (Lepidoptera: Pieridae), similar in color to the host plant, was taken as the object and the method for identifying and locating the target point was researched. A binocular camera unit with an optical filter of 850 nm wavelength was designed to capture the pest image. The segmentation of the pests' pixel area was performed based on Mask R-CNN. The laser strike points were located by extracting the skeleton through an improved ZS thinning algorithm. To obtain the 3D coordinates of the target point precisely, a multi-constrained matching method was adopted on the stereo rectification images and the subpixel target points in the images on the left and right were optimally matched through fitting the optimal parallax value. As the results of the field test showed, the average precision of the ResNet50-based Mask R-CNN was 94.24%. The maximum errors in the X-axis, the Y-axis, and the Z-axis were 0.98, 0.68, and 1.16 mm, respectively, when the working depth ranged between 400 and 600 mm. The research was supposed to provide technical support for robotic pest control in vegetables.

**Keywords:** robotic pest control; Mask R-CNN; skeleton extraction; binocular vision; stereo matching

## 1. Introduction

Physical pest control with laser power is widely considered as effective in reducing the pollution to the environment and even the damage to human health from the chemical pesticide [1,2]. Since 1980, many researchers have explored the outcome of pest elimination with lasers [3–5]. It has been demonstrated in these studies that laser power can cause damage to the exoskeleton and underlying tissues of pests, disrupt the anabolism of tissue cells, and ultimately kill pests [6,7]. Li et al. [5] found that the 24 h mortality rate of the fourth larval instar of *Pieris rapae* (L.) (Lepidoptera: Pieridae) reached 100% under the optimal working parameter combination of laser power of 7.5 W, an irradiation area of 6.189 mm$^2$, the laser opening time of 1.177 s, and the irradiation position in the middle of the abdomen. Therefore, to make laser pest control technology applicable in engineering settings, a pest control device is required to accurately focus the laser on the middle of the pest's abdomen to ensure that the laser kills the pests precisely under intense energy.

In this respect, machine vision technology can be applied to identify the pests present in the field [8,9]. However, most pests have a protective color for defense. In particular, the image background is complex and pest image features are less than prominent due to the intensive planting of crops [10]. Moreover, prior research on pest identification has mainly focused on the classification and counting of the pest species, with little attention paid to the 3D location of pests. Therefore, deep learning technology and binocular vision are integrated in this study to accurately identify and locate the laser strike point on the pest, thus providing technical support for robotic pest control in vegetables.

The mask regional convolutional neural network (Mask R-CNN) model first proposed by He et al. [11] can be used for instance segmentation and detection of pest images and achieves multiple research results in pest detection tasks [12,13]. Wang et al. [14] constructed a *Drosophila* instance segmentation model for automatically detecting and segmenting *Drosophila* wing, chest, and abdomen images, with an average precision of 94%. The instance segmentation can obtain target contour information without image morphological processing and is more suitable for accurate pest identification in laser pest control tasks. However, the above methods are used to segment RGB images of pests in specific environments, such as laboratory environments [15] and yellow sticky traps [16]. Existing algorithms still accurately segment pest targets with protective color characteristics in field environments.

As an extension of computer vision technology, near-infrared (NIR) imaging technology is used in insect species identification [17] and plant disease monitoring [18] widely. Sankaran et al. [19], based on visible-near infrared and thermal imaging technology, quickly identified citrus greening with an average precision of 87%. Luo et al. [20] used NIR imaging technology to track and monitor the structure and physiological phenology of Mediterranean tree-grass ecosystems under seasonal drought. Our team [21] proposed a monocular camera unit with an 850 nm optical bandpass filter to capture the image for identifying the pests, and the NIR image was confirmed to highlight the gray difference between the larvae of *P. rapae* and the vegetable leaves (Figure 1).

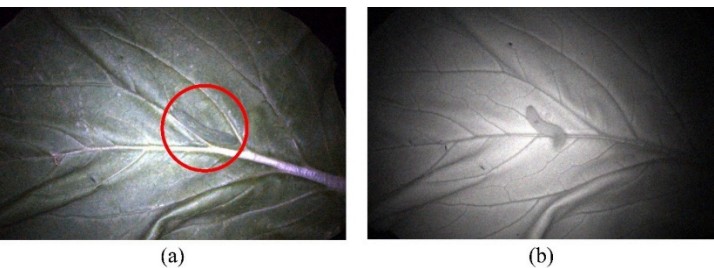

(a)  (b)

**Figure 1.** Comparison of near-infrared imaging effects of *Pieris rapae* on cabbage leaves. (**a**) The original image. (**b**) Near-infrared image. In the process of image acquisition, *P. rapae* and cabbage leaves were placed in a black box and an 850 nm infrared filter was installed on the camera to collect near-infrared images with an 850 nm ring light source. The original image is not equipped with a filter but is equipped with a white ring light with the same power as the 850 nm.

After identifying and segmenting pests in the field, the laser strike point is located in three dimensions based on binocular stereo vision. Stereo matching is an important factor affecting the location accuracy of binocular vision. Based on the constraint range and search strategy, the matching algorithm can be divided into local [22,23], global [24,25], and semi-global [26,27] stereo matching. However, the smaller larvae of *P. rapae* remain. With the 4th and 5th instar larvae of *P. rapae* as an example, their average widths reach 1.564 mm and 2.738 mm, respectively [28]. The above stereo matching of the global parallax map for the small target pests will result in low matching efficiency and poor location accuracy. Therefore, on the basis of the determined operation range, the candidate matching region was narrowed by the multi-constrained method to improve the efficiency and location accuracy of the stereo matching.

In this study, we designed a 3D locating system for pests' laser control to eliminate the above problems of inconspicuous pest image features, unclear location of strike points, and inefficient matching algorithms. A binocular camera unit with an optical filter of 850 nm wavelength was designed to capture the pest image. The ResNet50-based Mask R-CNN extracted the bounding box and the segmentation mask of the *P. rapae* pixel area, and the laser strike point was located in the middle of the pest abdomen, which was extracted through an improved ZS thinning algorithm with smoothing iterations. Furthermore, a multi-constrained matching method was adopted on the stereo rectification images. The

subpixel target points in the images on the left and right were optimally matched by fitting the optimal parallax value with the most similar feature between the template area among the two images. The 3D coordinates of each laser strike point were located according to its pixel coordinates in the two images. Finally, the recognition and localization performance of the system for targets at different locations was evaluated by implementing it on a field test platform. The research results can provide theoretical reference for the automatic laser strike of the pest control robot.

## 2. Materials and Methods

### 2.1. Binocular NIR Vision Unit

The 3D locating system was composed of a binocular vision system, a light source module, and host computer software, as shown in Figure 2a. In this system, the binocular vision system was composed of two gigabit industrial cameras produced by Hangzhou Haikang Robot Technology Co., Ltd. (Zhejiang, China). The camera model was MV-CA060-10GC, which is equipped with the lens model MVL-HF0628M-6MPE and a near-infrared filter of 850 nm. The resolution of each camera is 3072 (H) × 2048 (V), the focal length is 6 mm, and the frame rate is 15 fps. The two cameras were installed on the camera frame in parallel, and the baseline length was 50 mm. In addition, the system was illuminated by an 850 nm diffuse light bar, which can emit light evenly without shadows. The image processing platform adopted a Lenovo notebook ThinkPad P1, 24 GB RAM, Inter-Core i7-8750H@2.20 GHz, Windows 10, 64-bit system. The software system was mainly based on the OpenCV visual library and the TensorFlow deep learning framework.

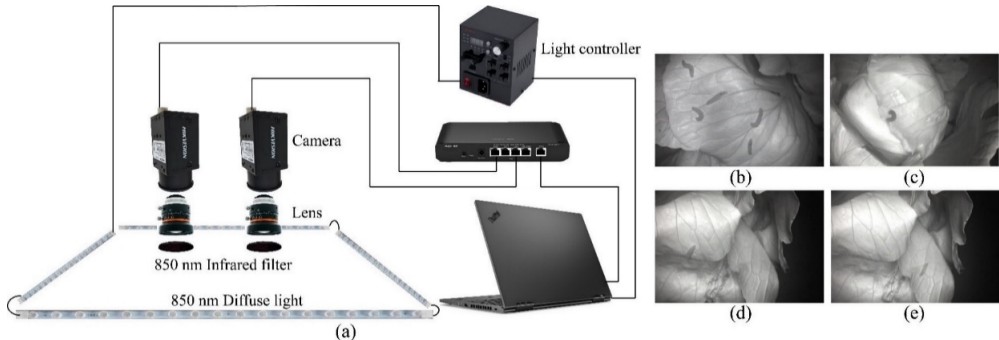

**Figure 2.** The binocular NIR vision unit and the example of collected images. (**a**) The visual system composition. (**b–e**) Regular cabbage as background and *P. rapae* larvae in different positions and postures taken from the collected NIR images.

Before image acquisition, a chessboard calibration board with a square size of 30 mm × 30 mm was used to perform stereo correction on the binocular camera [29]. In the process of image acquisition, the acquisition device was placed immediately above the cabbage leaves under natural illumination to collect images of *P. rapae* in the field. The collected images are shown in Figure 2b–e.

### 2.2. System Architecture

The flow of the field pest 3D locating system proposed in the study is shown in Figure 3, which mainly includes three parts: (1) pest identification and instance segmentation of the Mask R-CNN, (2) locating the laser strike point by extracting the skeleton of the pest, and (3) the 3D localization of laser strike point involved matching template preprocessing, multi-constraint narrowing of the matching region, subpixel stereo matching, and 3D coordinate extraction.

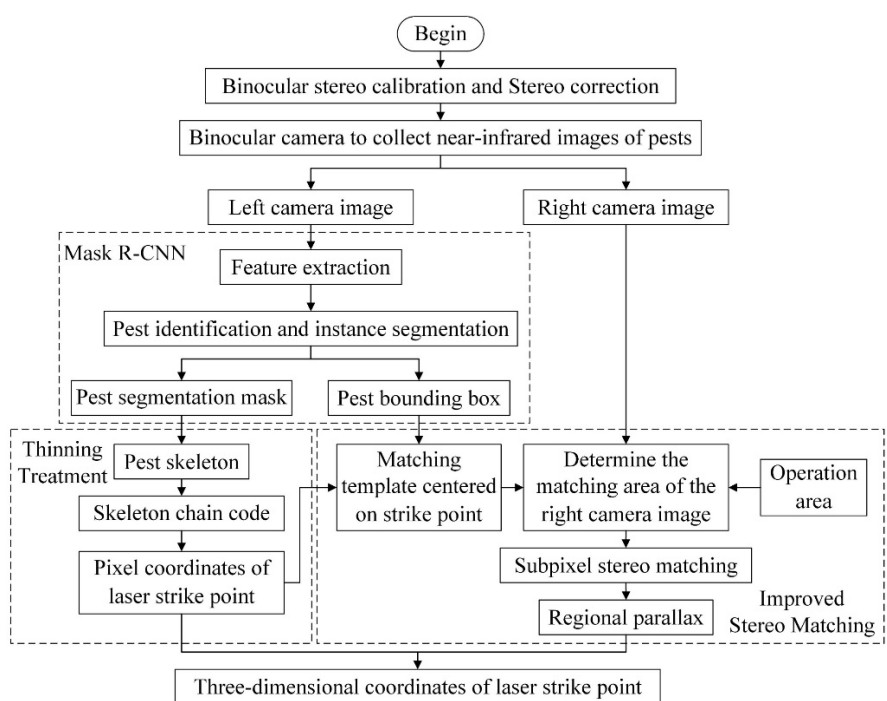

**Figure 3.** Structure diagram of the 3D locating system for field pests.

2.2.1. Instance Segmentation of *Pieris rapae* Image Area Based on Mask R-CNN

(1) Mask R-CNN Model

The accuracy of pest contour segmentation directly affects the accuracy of the laser strike point and stereo matching parallax. Based on the self-built NIR field *P. rapae* image dataset, this paper selected ResNet50-based Mask R-CNN [11] to identify and segment the pests' image area. The model structure is shown in Figure 4, which mainly includes the following steps:

1. The feature extraction network ResNet50 [30] extracted multi-scale information from the input image and generated a series of feature maps.
2. According to the mapping relationship between the feature map and the input image, the region proposal network (RPN) used the sliding window of the convolution layer to scan the anchor box in the feature map and generated a series of regions of interest (RoI) through classification and regression.
3. The RoI Align determined the eigenvalue of each point in the RoI and then performed pooling and other operations to match and align the target candidate region obtained by the RPN network with the feature map.
4. The feature maps output by RoI Align were input to the fully connected (FC) layers and the fully convolutional network (FCN). The former identified *P. rapae* and located the respective bounding boxes, and the latter segmented the pixel area of the larvae.

(2) Dataset augmentation and labeling

In total, 1000 images of *P. rapae* larvae in different poses were collected in the Brassica oleracea field. The sample numbers were expanded to 2000 by rotation, magnification, and horizontal and vertical mirroring, which improves the robustness of the recognition model [31]. Among them, each image contains at least one *P. rapae* larvae. We then marked the outline of *P. rapae* with the help of the open-source tool LabelMe. This tool can pick *P. rapae* masks from images and output a dataset in COCO format. Finally, the dataset was divided into a training set and a validation set according to the ratio of 8:2 for model training.

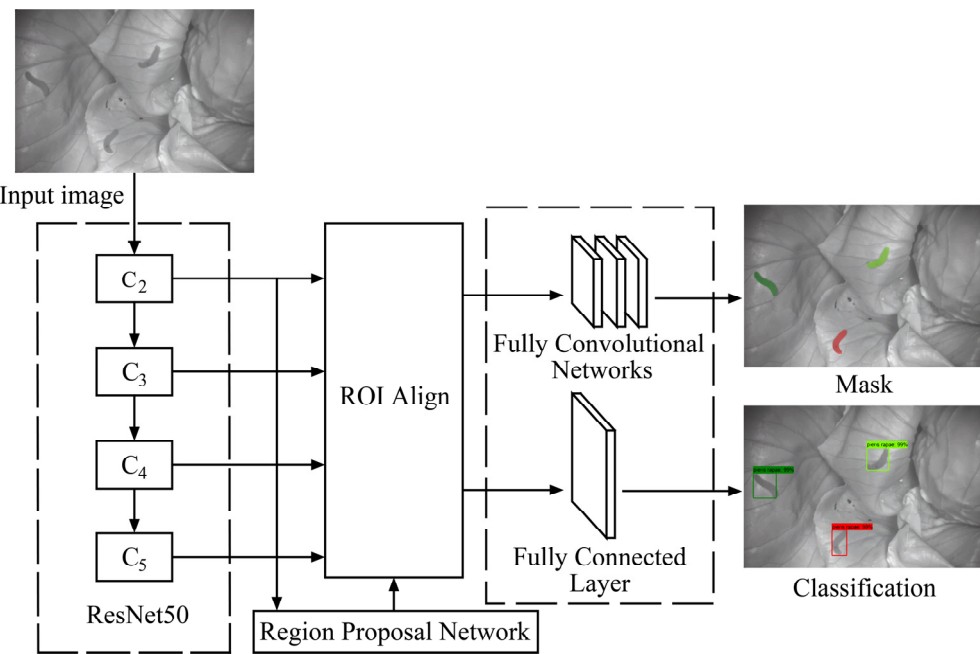

**Figure 4.** Overall Mask R-CNN with the ResNet50 model structure.

(3)    Transfer training

The model training was completed using a PC with the following hardware environment: 32 GB RAM, Inter-Xeon E5-2623 v3*2@3.00 GHz, and NVIDIA GeForce RTX2080. The software system uses the TensorFlow deep learning framework under Windows 10 and 64-bit operating systems for coding and training and was configured with Python3.6, Anaconda 5.3.1, and CUDA10.0 compilation environments.

The training method adopted the transfer training method. The Mask R-CNN was initialized with the feature extraction network weights of the pre-trained model, while the object classification, bounding box regression, and FCN parameters were randomly initialized. During training, the initial learning rate was 0.001, the momentum parameter was 0.9, and the batch size was set to 1. In the RPN structure, the anchor point sizes were 32, 64, 128, 256, and 512. The anchor point frame ratio was 0.5:1:2.

The model object detection and region segmentation results are shown in Figure 5. The high-quality segmentation mask distinguishes pests from the background, which can be used to calculate the location of the laser strike point directly.

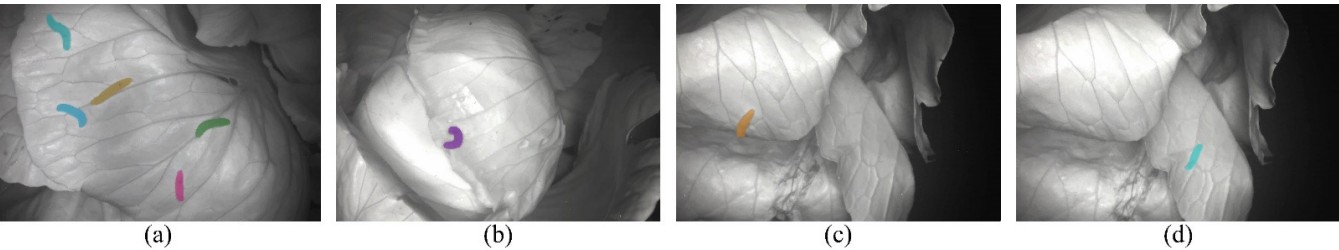

(a)                          (b)                          (c)                          (d)

**Figure 5.** Visualization results of the ResNet50-based Mask R-CNN. (**a**–**d**) *P. rapae* larvae in different positions and postures taken from the collected NIR images. (**a**) Multiple pests, (**b**) curled pests, (**c**) occlusion state, and (**d**) dorsal position of the leaf.

### 2.2.2. Pest Skeleton Extraction and Strike Point Location

(1)    Laser strike point

Laser pest control requires focusing the laser on the middle of the pest abdomen to ensure that the laser kills the pests with intense energy. The body of *P. rapae* larvae is

tubular and segmented, as shown in Figure 6. The middle part of the abdomen irradiation position was between the 8th and 9th segments, near the midpoint of the skeleton [5,32]. Therefore, this paper set the laser strike point as the midpoint of the skeleton of the pest image area. The improved ZS thinning algorithm was used to extract pest skeletons. Then, pest skeleton chain code was established to extract the skeleton midpoint coordinates to determine the final strike point.

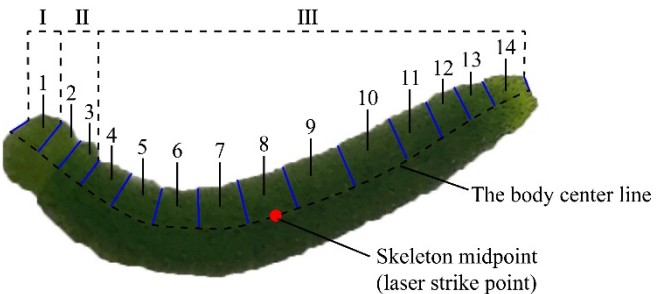

**Figure 6.** Characteristics of the *Pieris rapae* larvae and locating the laser strike point. The body of *P. rapae* larvae can be divided into the head (I), the thorax (II), and the abdomen (III). The numbers 1–14 denote the different segments of the larvae, separated by blue lines.

(2)    Pest skeleton extraction based on improved ZS thinning algorithm

The skeleton consists of a single pixel, which provides an orientation for extracting the laser strike point coordinates. However, due to the different positions and postures of pests in the field and the sensitivity of the traditional skeleton extraction algorithm to the boundary, the extracted pest skeletons display the phenomenon of a non-single-pixel width and end branches, as shown in Figure 7.

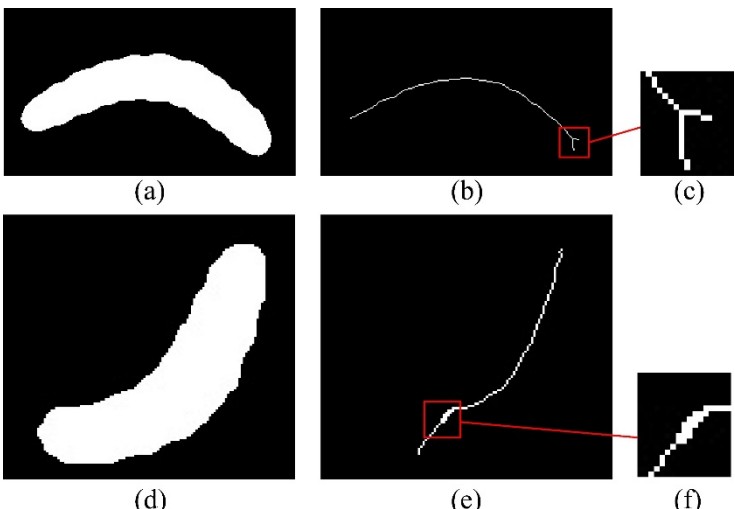

**Figure 7.** Visualization of the ZS thinning algorithm. (**a**,**d**) The segmentation mask of the Mask R-CNN. (**b**,**e**) The pest skeleton images. (**c**,**f**) The local details of the pest skeleton.

To solve these problems in the above-mentioned thinning process, this paper introduced an improved ZS thinning algorithm [33] with smoothing iterations to extract pest skeletons. The whole skeleton process was divided into three iterative processes: smooth iteration, global iteration, and two-stage scanning.

In the smooth iteration, the candidate deletion points were extracted based on the refinement constraints of the traditional ZS algorithm. Then, the smooth pixel points in the candidate deletion points were preserved in the smooth iteration process, which suppress

the branching at the end of the pest skeleton, as shown in Figure 8. Among them, the definition of smooth pixel points satisfies Equation (1):

$$5 \leq N_b(P_0) \leq 6 \tag{1}$$

where $N_b(P_0)$ denotes the number of pixels with value 1 in the neighborhood of the scanning point $P_0$.

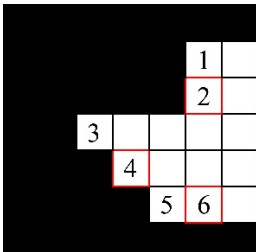

**Figure 8.** Example of the smooth pixel point determination. The numbers 1–6 denote the candidate deletion points extracted by the ZS thinning algorithm, where 2, 4, and 6 denote the smooth pixel points.

In smoothing iteration and global iteration, the reserved template under 24 neighborhood subdomains was added. The candidate deletion points that meet the retention template were reserved, which avoided the problem of topological structure deletion. Figure 9a–i shows the pixel set of the retention templates. The 24 neighborhood pixels were divided into 4 × 4 subdomains in 4 different directions for generating specific structures in different directions. Figure 9a–h was used to maintain diagonal lines of two-pixel widths, and Figure 9i was used to maintain the 2 × 2 square structure.

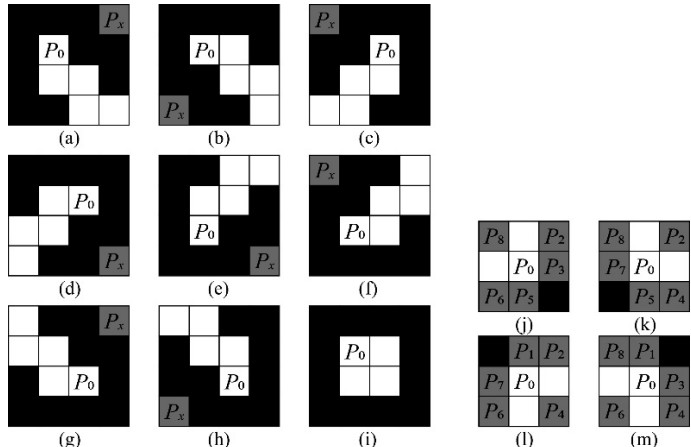

**Figure 9.** The retention templates and the deletion templates. (**a–i**) The retention templates in different directions. (**j–m**) The deletion templates in different directions. The pixels of scanning points are marked as $P_0$, and pixel sets $P_x$ of 8 neighborhoods and 24 neighborhoods are constructed, where $x = 1, 2, \ldots 24$. The pixel $P_x$ in the gray square can be either 1 or 0.

In the two-stage scanning, the deletion templates under 8 neighborhoods were used to eliminate the pixels with non-single-pixel widths that form an included angle of 90. The definition of the deletion templates satisfied Figure 9j–m.

Based on the improved ZS thinning algorithm, the pest skeletons in Figure 7a,d were extracted again. The visualization is shown in Figure 10.

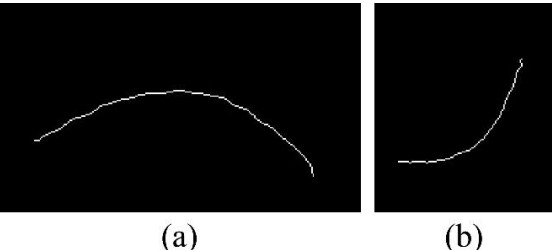

**Figure 10.** Visualization of the improved ZS thinning algorithm. (**a**,**b**) The pest skeleton images extracted from Figure 7a,d.

(3)    Strike point location

After extracting the skeleton of pests with a single-pixel width, the system used Freeman chain code notation [34] to extract the linked list. Then, the skeleton pixel length was calculated by combining the chain code and the midpoint position coordinate was located according to the pointer. The visualization results of different processing stages are shown in Figure 11.

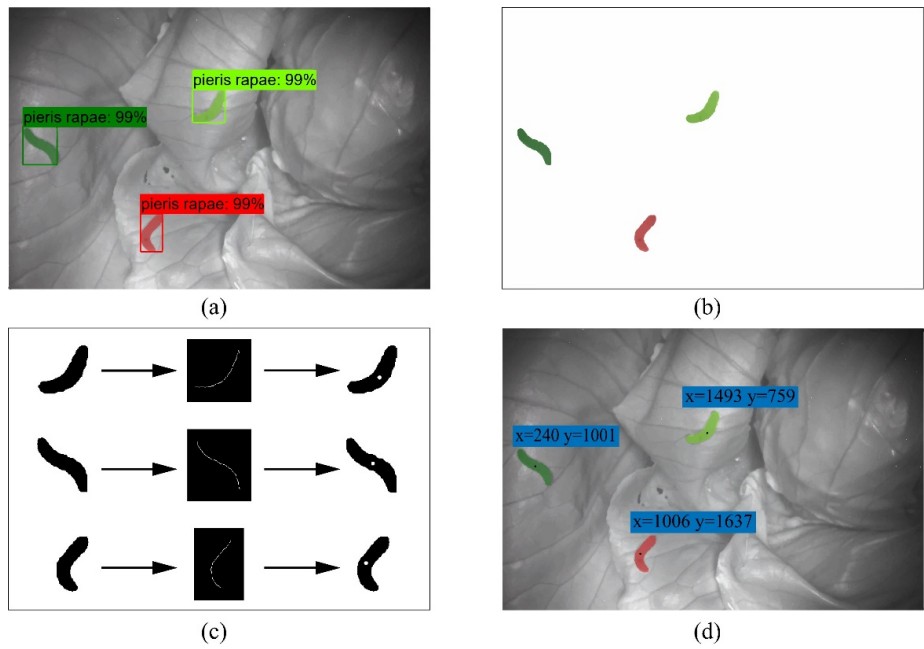

**Figure 11.** Visualization of the pest skeleton extraction and laser strike point location for different stages: (**a**) The identification and segmentation result of an NIR *P. rapae* image, (**b**) extracted segmentation mask image, (**c**) thinning treatment, and (**d**) coordinates of laser strike points.

### 2.2.3. The Multi-Constrained Stereo Matching Method

In this study, we only need to calculate the 3D spatial coordinates of the laser strike point and, thus, a multi-constraint stereo matching algorithm was proposed. As shown in Figure 12, the algorithm constructs two constraints in the matching process.

(1)    The first construct: Row Constraint

After the binocular camera (Figure 12a) completed the camera calibration and stereo correction, the same pest satisfied the constraint of peer-to-peer sequential consistency in the stereo rectification images [35]. Therefore, using the pest segmentation mask in the image on the left as the template, template matching was performed on the same row in the image on the right according to the row constraint.

Assuming that the coordinate of the laser strike point in the image on the left was $p_1(x_1, y_1)$, the range of the coordinate $p_2(x_2, y_2)$ of the center point of the matching box in the image on the right can be limited to $y_2 = y_1$, as shown in Figure 12b.

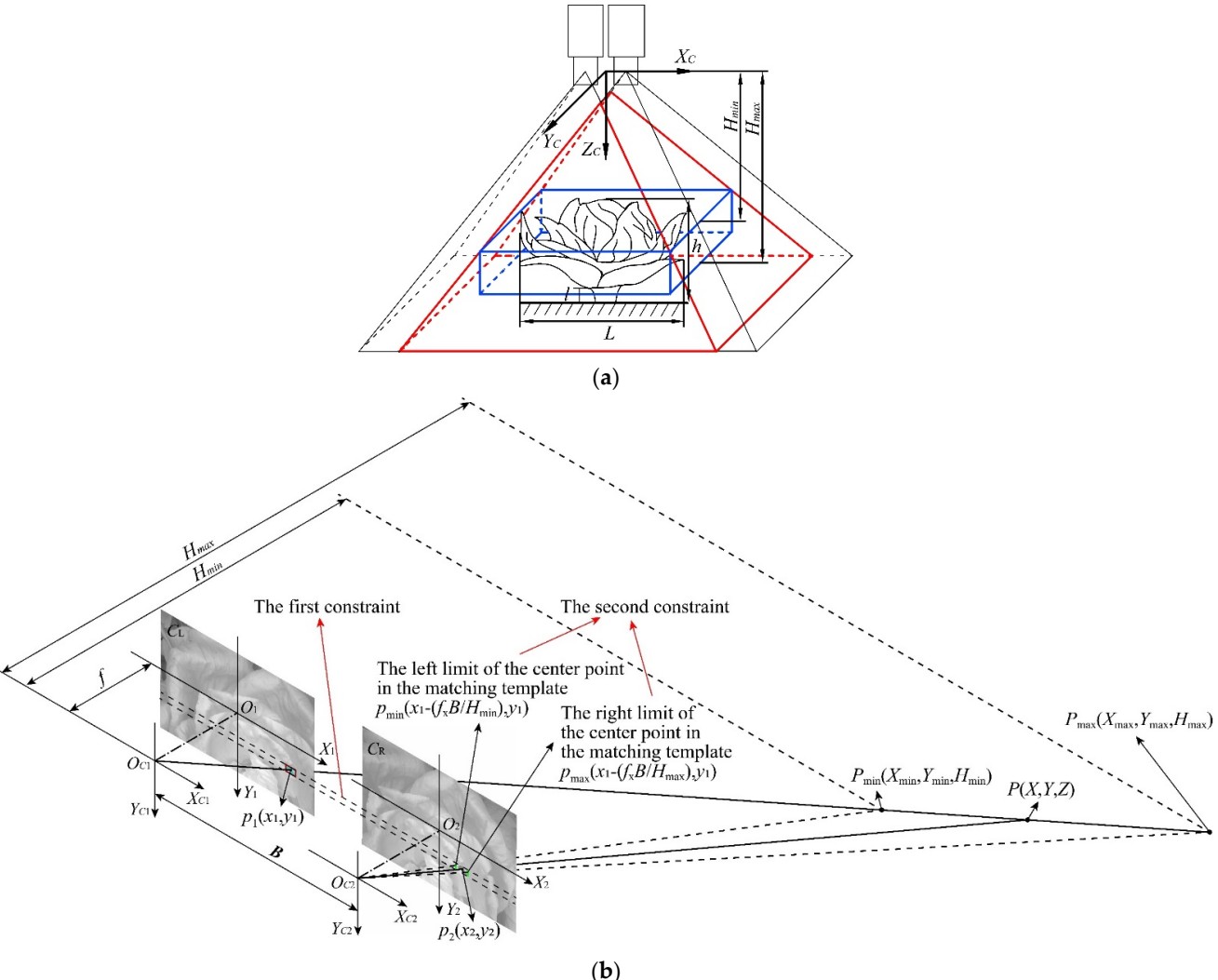

**Figure 12.** Search range of the multi-constraint stereo matching method. (**a**) The binocular vision locating system. The red frame is the binocular public area, the blue frame is the operation area for locating pests, and the depth range is $H_{min} \sim H_{max}$. $L$ is the leaf spreading degree; $h$ is the plant height; and $l$ is the bottom leaf height of the cabbage. $f_x$ is the camera fixed parameter. (**b**) The spatial geometric diagram. $O_{C1}$ and $O_{C2}$ are the optical centers of the cameras on the left and right, $C_L$ and $C_R$ are the imaging planes of the binocular cameras, and the image coordinate systems are $X_1O_1Y_1$ and $X_2O_2Y_2$, respectively. $p_1(x_1, y_1)$ is the laser strike point in the image on the left; $p_2(x_2, y_2)$ is the center point of the best matching box in the image on the right; $P(X, Y, Z)$ are the target pests.

(2) The second construct: Column Constraint

For the laser pest control robot to effectively identify field pests and facilitate the trajectory planning of its striking equipment, the working area was regarded as a cuboid (Figure 12a). According to the principle of triangulation [35], the coordinate of the target point in the world coordinate system can be calculated by Equation (2):

$$Z = \frac{fB}{(x_1 - x_2)\mu_x} = \frac{f_x B}{x_1 - x_2} \tag{2}$$

where $B$ is the baseline distance of the binocular cameras, $f$ is the focal length of the cameras, $\mu_x$ is the physical size of each pixel in the $X$-axis direction of the imaging plane, and $f_x$ is the fixed parameter of the camera, which is determined during camera calibration.

In Equation (2), if the depth range of the operation area, the coordinate $p_1(x_1, y_1)$ of the target in the image on the left, and the camera fixed parameter $f_x$ were known, the range of the $X$-axis of the target in the image on the right can be limited. The specific equation of $x_2$ was as follows.

$$x_1 - \frac{f_x B}{H_{min}} \leq x_2 \leq x_1 - \frac{f_x B}{H_{max}} \tag{3}$$

where $H_{min}$ and $H_{max}$ are the value ranges of the $Z$-axis of the system operation area in the world coordinate system (Figure 12).

Based on the multiple constraints above, the matching range of the template on the polar line of the target image on the right can be further restricted.

In the matching process, the normalized cross-correlation coefficient with linear illumination invariance was selected to measure the match similarity [36]:

$$R(x,y,d) = \frac{\sum_{i=1}^{n} \sum_{j=1}^{m} \left[ T(x+i, y+j) - \overline{T}(x,y) \right] \left[ I(x+i-d, y+j) - \overline{I}(x-d,y) \right]}{\sqrt{\sum_{i=1}^{n} \sum_{j=1}^{m} \left[ T(x+i, y+j) - \overline{T}(x,y) \right]^2} \sqrt{\sum_{i=1}^{n} \sum_{j=1}^{m} \left[ I(x+i-d, y+j) - \overline{I}(x-d,y) \right]^2}}, d \in \left[ \frac{f_x B}{H_{max}}, \frac{f_x B}{H_{min}} \right] \tag{4}$$

where $R(x,y,d)$ is the normalized correlation quantity when the midpoint $(x,y)$ is located in parallax $d$ in the matching area of the camera image on the right. Here, $n$ is the width of the template window; $m$ is the height of the template window; $T(x+i, y+j)$ is the pixel value of the template window point $(x+i, y+j)$; and $\overline{T}(x,y)$ is the average pixel value of the template window. $I(x+i-d, y+j)$ is the pixel value of the matching area point $(x+i-d, y+j)$; and $\overline{I}(x-d,y)$ is the average pixel value of a template window with a side length of $m \times n$ defined by the point $(x-d, y)$ as the center.

After obtaining the parallax $d_0$ with the maximum similarity (Equation (4)), the algorithm extracted the matching similarity $R(x,y,d)$ of the adjacent parallaxes ($d_0 - 2, d_0 - 1, d_0 + 1, d_0 + 2$) with phase-pixel-level accuracy and constructed a parallax-similarity ($d$-$R$) pointset, as shown in Figure 13. Then, the quadratic, cubic, and quartic polynomial fitting curves were performed on the pointset to obtain the polynomial curve with the highest fitting degree ($R^2$). The abscissa of the crest (Figure 13, Point $S$) at the best fitting curve was the parallax under subpixel accuracy. Finally, the 3D coordinates of each pest in the world coordinate system were calculated by the subpixel parallax.

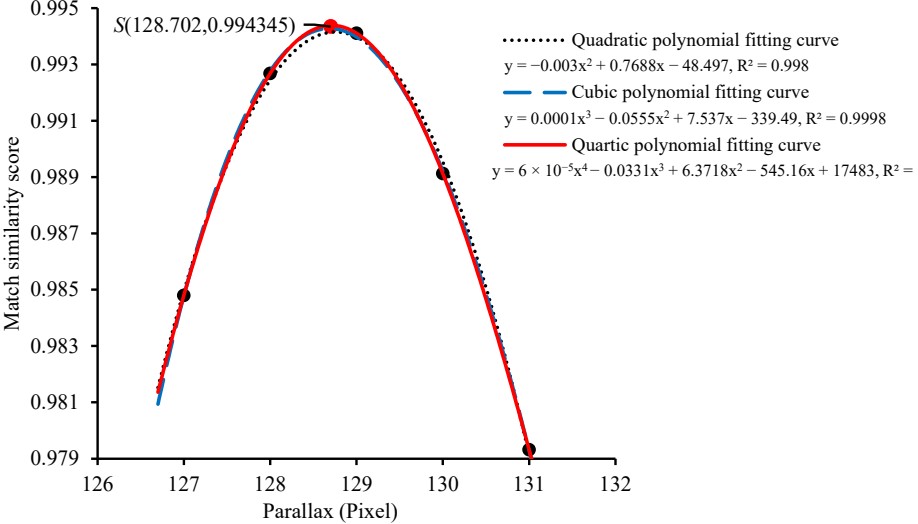

**Figure 13.** Polynomial fitting curves of disparity and similarity.

## 3. Test and Results

### 3.1. Experiments

To evaluate the recognition and localization accuracy of the laser strike point, combined with the characteristics of the actual operating conditions of the cabbage greenhouse, we further collected the *P. rapae* images at different positions in the vegetable field to construct a test set (Experiment 1: $n = 70$, Experiment 2: $n = 30$). The system automatically outputs and saves the identification and segmentation results of the *P. rapae* pixel area and records the 3D coordinates of the laser strike point.

The experiment was conducted in the cabbage field (28.18 N, 113.07 E) of Hunan Agricultural University in Changsha, Hunan Province, as shown in Figure 14. According to the leaf spreading degree ($350 \pm 46.6$ mm), plant height ($300 \pm 25.6$ mm), and bottom leaf height ($32 \pm 6.7$ mm) of the field cabbage, the distance between the origin of the binocular camera and the effective operation area of the laser was set to 400–600 mm. The length of the working area along the $X_C$-axis was 400 mm and the $Y_C$-axis was 260 mm.

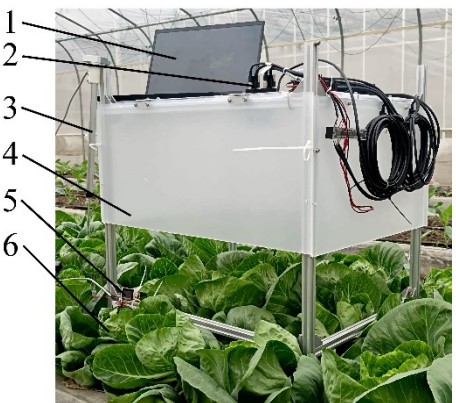

**Figure 14.** Accuracy test platform site. Key: 1. visual processing platform; 2. binocular camera with an 850 nm filter; 3. linear displacement sensor; 4. fixed support frame; 5. digital display for displacement sensor; 6. cabbage.

3.1.1. Experiment 1: Accuracy Evaluation of Pest Identification and Instance Segmentation Network

Combined with the test sample images ($n = 70$) of different scenarios, the number of *P. rapae* that were manually labeled and automatically identified by the model were recorded. Three indicators, precision value (Equation (5)), recall value (Equation (6)), and $F_1$-measure (Equation (7)), were used to evaluate the recognition performance of the Mask R-CNN model on the target.

$$P = \frac{TP}{TP + FP} \tag{5}$$

$$R = \frac{TP}{TP + FN} \tag{6}$$

$$F_1 = \frac{2 \times PR}{P + R} \tag{7}$$

where *TP* is a correctly predicted positive sample, *FP* is an incorrectly predicted negative sample, and FN is an incorrectly predicted positive sample.

3.1.2. Experiment 2: Performance Evaluation of the 3D Locating System

The image coordinate deviation and the actual depth deviation between the auto-location results of the laser strike point and the manual annotation results were used to evaluate the performance of the 3D locating system.

Given that the absolute deviation of coordinates represents different physical distances in images of different scales, it is impossible to characterize the true locating error

quantitatively. In experiment 2, we collected 30 pairs of binocular images of the same *P. rapae* at different locations in the vegetable field. Therefore, it is assumed that the physical diameter of the *P. rapae* body width in the area of the laser strike point was constant and *d* represented the pixel width of *P. rapae* body in images of different scales (Figure 6). The *X*-axis, *Y*-axis location error of the world coordinate system was represented by the ratio of the pixel deviations ($e_x$, $e_y$) and *d* of the system output and the manually marked point on the *x* coordinate, *y* coordinate of the image.

In experiment 2, a linear displacement sensor (provided by Shenzhen Howell Technology Co., Ltd. (Shenzhen, China), KPM18-255) was used to measure the vertical distance from the pest surface to the camera plane. The sensor position accuracy was 0.05 mm. The displacement sensor is installed in a base with a magnet. The base can be adsorbed on the top plate in such a way that the displacement sensor is always perpendicular to the imaging plane and can move horizontally in the plane of the top plate, as shown in Figure 15.

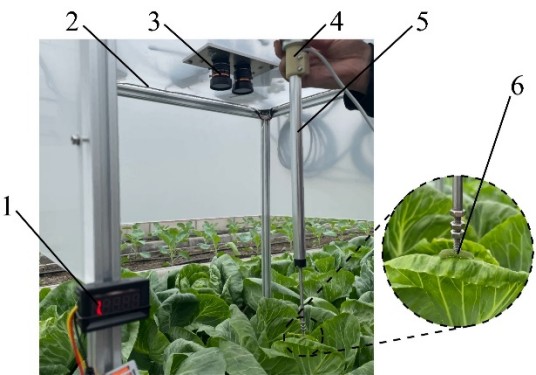

**Figure 15.** Accuracy testing experiment equipment. Key: 1. digital display for displacement sensor; 2. 850 nm diffuse light bar; 3. binocular camera with an 850 nm filter; 4. base with a magnet; 5. linear displacement sensor; 6. *Pieris rapae*.

### 3.2. Validity Results of Mask R-CNN

The model training (Section 2.2.1) results showed that the average precision (AP), $AP^{0.50}$, and $AP^{0.75}$ of the ResNet50-based Mask R-CNN model constructed on the self-built NIR field *P. rapae* image dataset reached 94.24%, 98.74%, and 96.79%, respectively.

Manual detection was performed on 70 images in the test set. The target distribution of the test set was actually 158 *P. rapae* larvae, and each image contains at least one.

Then, the test set images were input into the above models. The object detection results of the larvae in the image samples of the test set by the model are shown in Table 1. The values of precision, recall, and $F_1$ were 96.65%, 97.47%, and 96.55%, respectively, showing the effectiveness of the proposed model.

**Table 1.** Identification results for the *P. rapae* larvae in the test set.

| Number [1] | | | | Precision (%) | Recall (%) | $F_1$ (%) |
|---|---|---|---|---|---|---|
| *N* | *TP* | *FP* | *FN* | | | |
| 158 | 154 | 3 | 4 | 95.65 | 97.47 | 96.55 |

[1] *N* is the total number of larvae in the test set. *TP*, *FP*, and *FN* are the quantities of correctly predicted positive samples, incorrectly predicted negative samples, and incorrectly predicted positive samples, respectively.

### 3.3. 3D Localization Results of Field Pests

The binocular stereo vision system completed the camera calibration and stereo correction, and the results are shown in Table 2. The reprojection error was 0.36 pixels, and the calibration results meet the test requirements [37].

**Table 2.** The internal and external parameters of the binocular stereo vision system.

| Parameters | Left Camera | Right Camera |
|---|---|---|
| Focus/mm | 6 | |
| Cell size/μm | 2.4 (Sx) × 2.4 (Sy) | |
| Center column (Cx)/pixel | 1589.60 | 1609.84 |
| Center row (Cy)/pixel | 1034.15 | 1051.87 |
| 2nd order radial distortion (K1)/1/pixel$^2$ | −0.087540 | −0.086044 |
| 4th order radial distortion (K2)/1/pixel$^4$ | 0.162294 | 0.155954 |
| 6th order radial distortion (K3)/1/pixel$^6$ | 0.000185 | 0.000337 |
| 2nd order tangential distortion (P1)/1/pixel$^2$ | 0.000210 | −0.000308 |
| 2nd order tangential distortion (P2)/1/pixel$^2$ | −0.065631 | −0.056233 |
| Image size/pixel | 3072(H) × 2048(V) | |
| Baseline distance/mm | 49.50 | |
| Reprojection error/pixel | 0.36 | |

### 3.3.1. *X*-Axis and *Y*-Axis Location Error

In this paper, the ratio of the image positioning deviation of the laser strike point of different scales to the pixel width of the *P. rapae* body was used as the *X*-axis and *Y*-axis location error, and the results are shown in Figure 16. In the sample images of the whole test set (N = 30), all larvae were correctly recognized and segmented and the average image location errors in the *x* coordinate and the *y* coordinate of the laser strike point were 0.09 and 0.07, respectively. The maximum errors in different scenarios were 0.23 and 0.16.

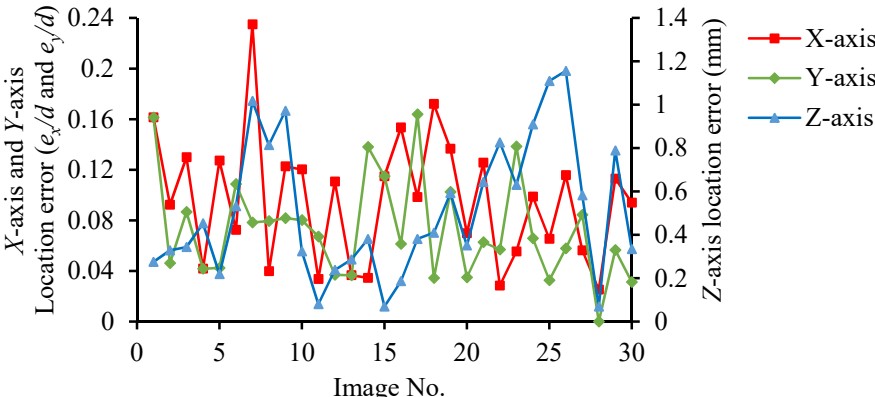

**Figure 16.** The location error of the laser strike point on the *X*-axis, the *Y*-axis, and the *Z*-axis. *d* denotes the pixel width of the *P. rapae* body in images of different scales. The location error is represented by the ratio of the x coordinate, y coordinate deviation ($e_x$,$e_y$) and *d*.

In the experiment, the same *P. rapae* larvae were used in different locations of the vegetable field and the larval body width was 4.16 mm (Manual measurement). Therefore, the average absolute error of the *X*-axis of the laser strike point was 0.40 mm and the maximum error was 0.98 mm. The average absolute error of the *Y*-axis was 0.30 mm, and the maximum error was 0.68 mm.

Considering the distance between the real and the located point, the average absolute error of the total location error in the *X*–*Y* plane was 0.53 mm, and the maximum error was 1.03 mm. All the located point were within the effective strike range in the middle of the pest abdomen (Figure 6).

### 3.3.2. *Z*-Axis Location Error

Analysis of Figure 16 shows the visual location error in the depth direction of the system when the working depth was between 400 and 600 mm. The average absolute error was 0.51 mm, and the maximum error value was 1.15 mm. The root mean square error and the mean absolute percentage error of the system were 0.58 mm and 0.10%, respectively,

which shows that there is a strong explicit correlation between the estimated depth and the actual depth of the system.

## 4. Discussion

An automatic laser strike point localization system was established in this study based on the multi-constraint stereo matching method, which provided a basis for pests' laser control. Three aspects of the proposed model will be discussed in this section, i.e., the effects of the segmentation model, the effect of the location method, and the effect of the stereo matching method. Further improvements for the 3D locating system will also be pointed out in this section.

### 4.1. Analyses of Instance Segmentation Result

Experiment 1 showed that the segmentation results (AP, $AP^{0.50}$, and $AP^{0.75}$) of the ResNet50-based Mask R-CNN model were higher than 94% on the self-built NIR image dataset of *P. rapae*. The good segmentation performance of the network proves that the application of near-infrared imaging technology is feasible for pest identification, with protective color characteristics in multi-interference scenes.

In the sample images of the whole test set, the number of correctly predicted, incorrectly predicted, and unrecognized *P. rapae* were 154, 3, and 4, respectively. Among them, the number of incorrectly predicted and unrecognized *P. rapae* in a single *P. rapae* image was 0. The main causes of errors are: (1) When two or more *P. rapae* larvae overlap each other, the larvae bodies are blocked. This situation increases the difficulty of identification, resulting in multiple pests being identified as a whole or a single pest being only partially segmented (Figure 17a). (2) In the near-infrared image, the soil color is close to that of the cabbage bugs. When a leaf has a hole to expose the soil and the shape is a long strip, the model will misjudge it as a *P. rapae* larva (Figure 17b). Furthermore, the complicated network structure also makes the training time of Mask R-CNN longer. The detection time for a single image in the segmentation network was 460 ms.

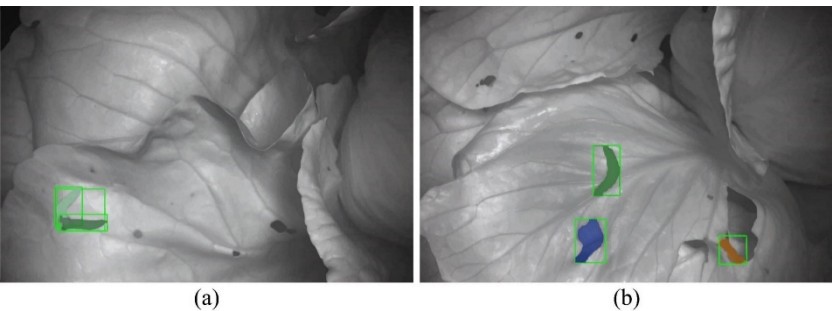

(a)          (b)

**Figure 17.** False identification results. (**a**) Two *P. rapae* larvae overlap each other and (**b**) leaf holes mistakenly identified as *P. rapae*.

### 4.2. Analyses of Location Result

According to the segmentation mask in the bounding box, the laser strike point was located as the midpoint of the skeleton of pest image area, which was extracted through an improved ZS thinning algorithm. This method solves the problem of pest contour extraction based on deep learning, which greatly improves the robustness and efficiency of the algorithm.

However, this method cannot accurately locate the laser strike point in some special cases. The main causes of errors are: (1) When the *P. rapae* is partially occluded by leaves or the inclination angle is large, the method of locating the laser strike point through the midpoint of the skeleton is inaccurate because only a part of the pest skeleton is extracted (Figure 18a). (2) If the *P. rapae* larvae curl up in a ring, the pest segmentation mask is a circle. The laser strike points finally obtained by the above location method is near the center of the circle and is not within the effective strike range (Figure 18b).

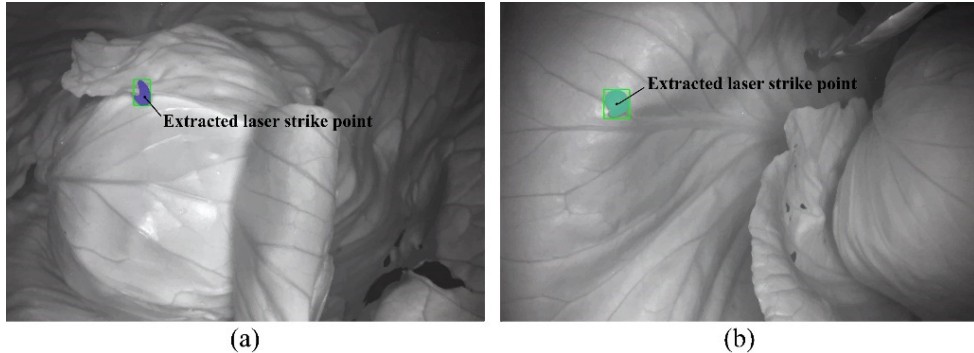

(a)        (b)

**Figure 18.** Incorrect location results in special cases. (**a**) The body of the *P. rapae* shaded by leaves. (**b**) The larvae curl up in a ring.

Fortunately, the above situation is not common. Fieldwork indicates that the *P. rapae* larvae are mostly found on the leaf surface in the morning, sunset, and night and are mainly located on the petioles, leaf veins, and undeveloped new leaves of the outer leaves. Especially at sunrise and at night, the *P. rapae* larvae can be clearly seen from the top of the plant when illuminated with light. The larvae curl up only when hit by external stimuli and usually become strip shaped. In general, the location method is suitable in most cases. However, the method still needs to be further improved to adapt to complex working conditions.

### 4.3. Analyses of the Multi-Constraint Stereo Matching Result

Experiment 2 showed that the average location errors on the *X*-axis, the *Y*-axis, and the *Z*-axis of the laser strike point were 0.40, 0.30, and 0.51 mm, respectively, and the maximum errors were 0.98, 0.68, and 1.16 mm. The system has high location accuracy on the *X*-axis and the *Y*-axis. Considering the distance between the real and the located point, the average absolute error of the total location error in the world coordinate system was 0.77 mm. The maximum error was 1.45 mm.

With the fourth and fifth instar larvae of *P. rapae* as an example, their average widths reach 1.564 mm and 2.738 mm, respectively [28]. Considering that the laser strikes vertically downward and the irradiation area is 6.189 mm$^2$ (diameter 2.8 mm) [5], the effective stroke of the laser end effector is increased by a maximum of 1.45 mm for accommodating the location error of the laser strike point. The extra travel poses less technical risk to the design and motion control of the laser strike device. The results satisfy the localization requirements of lasers to strike *P. rapae* larvae accurately.

The reasons for the errors are as follows: As the depth increases, the proportion of the pest area in the whole image is smaller, which results in pest segmentation and location errors. There are errors in internal and external parameters, which lead to an increase in the system error. Moreover, manual measurement error of the displacement sensor can also result in errors.

Overall, the average time of the entire pest localization process, including field pest identification, contour segmentation, and 3D coordinate position, was 0.607 s. Because the matching area was reduced, the stereo matching algorithm proposed in the study takes only 24.2% of the total time, approximately 0.147 s, which shows that the matching algorithm can quickly and accurately locate the three-dimensional coordinates of pests in the field after obtaining the pest segmentation results.

### 4.4. Discussion about Further Improvement Aspects

The data for this experiment were mainly collected at a depth of 400–600 mm above the ground. In the follow-up research, the relationship between the spatial resolution of the image and the laser strike point location accuracy of the proposed system can be further analyzed to obtain the best spatial solution. In this experiment, all images were

collected from directly above. However, this will result in a lack of image information for pests that may be occluded by leaves or have a larger body inclination. This is somewhat detrimental to understanding the overall situation of pest infestation. In future research, the data of pests located on leaves should be collected from multiple angles to generate well-established and accurate 3D location information of pests.

### 5. Conclusions

A novel 3D locating system based on binocular vision was proposed for laser pest control, combining a Mask R-CNN, pest skeleton extraction, and multi-constraint stereo matching. The ResNet50-based Mask R-CNN model was trained and validated with a self-built NIR field *P. rapae* image dataset collected in a real-world agriculture scene. The AP, recall, and $F_1$ values were 94.24%, 97.47%, and 96.55% of the Mask R-CNN, respectively, showing the adaptability of the proposed model.

Furthermore, when the working depth varied between 400 and 600 mm, the average location errors were 0.40 mm, 0.30 mm, and 0.51 mm and the maximum errors were 0.98, 0.68, and 1.16 mm for the 3D system in the *X*-axis, *Y*-axis, and *Z*-axis direction. The conclusions of this study provide a design basis for the follow-up research and development of the laser pest control execution system.

Since the laser strike point extraction in this paper was limited to the processing of two-dimensional image features, there is still room for improvement in object point localization methods and accuracy evaluation experiments. In the future, the depth camera can be further used to obtain the overall 3D pose information of the pests to improve the target localization accuracy.

**Author Contributions:** Conceptualization, Y.L., Y.X. and Q.F.; methodology, Y.L., Y.X. and Q.F.; software, Y.L.; validation, Y.L., J.L., X.L. and Z.H.; formal analysis, Y.L.; investigation, Y.L., J.L., X.L. and Z.H.; resources, Y.X. and Q.F.; data curation, Y.L.; writing—original draft preparation, Y.L.; writing—review and editing, Y.X. and Q.F.; visualization, Y.L.; supervision, Y.X. and X.L.; project administration, Y.X.; funding acquisition, Y.X. and Q.F. All authors have read and agreed to the published version of the manuscript.

**Funding:** This research was funded by the National Key Research and Development Plan Project (2019YFE0125200), the Natural Science Foundation of Hunan Province of China (2021JJ30363), the BAAFS Innovation Capacity Building Project (KJCX20210414), and the Science and Technology General Project of Beijing Municipal Education Commission (KM202112448001).

**Institutional Review Board Statement:** Not applicable.

**Informed Consent Statement:** Not applicable.

**Data Availability Statement:** All data are presented in this article in the form of figures and tables.

**Conflicts of Interest:** The authors declare no conflict of interest.

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
