# Peer review of "3D Locating System for Pests’ Laser Control Based on Multi-Constraint Stereo Matching"

_agriculture, doi:10.3390/agriculture12060766_

Round 1

Reviewer 1 Report

I suggest minor improvement in language. If possible reduce the number of figures because too many figures will not exactly convey the message to the audience. Figure 17 and Figure can be combined and presented as table (mean and SE).    

Author Response

Dear Reviewer:

Thank you very much for your comments concerning our manuscript entitled ‘3D Locating System for Pests’ Laser Control Based on Multi-constraint Stereo Matching’ (ID: agriculture-1700974). Those comments (at the bottom of the cover letter) are all valuable and very helpful for revising and improving our paper, as well as the important guiding significance to our researches. We have studied comments carefully and have made correction which we hope meet with approval.

Since the cover letter contains figures and tables, please see the attachment.

The above are my main corrections in the paper and the responds to the Reviewer’s comments. Special thanks to you for your comments. We tried our best to improve the manuscript and made some changes in the manuscript. Any revised portion made to the manuscript are marked up using the “Track Changes” function. At the same time, we appreciate for Editors and Reviewers warm work earnestly, and hope that the correction will meet with approval.

Once again, thank you very much for your comments and suggestion.

Yours sincerely,

Yajun Li

Reviewer 2 Report

The Manuscript entitled (3D locating system for pests’ laser control based on multi-constraint stereo matching) provide a novel 3D location system based on the binocular vision for pests’ laser control. The manuscript has many data and good results those are introduced very well in all sections.

Here, some comments for the authors that are considered as minor revision

Line 33: (of 4th larval instar of Pieris rapae)  instead of (of 4th instar Pieris rapae)

Line 50: Drosophila to be in italic and in all the manuscript such as line 51

Line 119: (Figure 2b,c.d and d) instead of (Figure 2b)

Line 180: Indicate in the figure legend what are a,b,c and d.

In figure 5: all words in the figure are not obvious, they should be with higher font.

Line 236: Indicate in the figure legend what are a and b.

Figure 11: all words in the figure  (a and d) are not obvious, they should be with higher font

Figure 16: Indicate in the figure legend what are a and b.

Line 471: The

Conclusion: It is very long with details. Please summarize this section with only valuable conclusion and recommendations

Author Response

Dear Reviewer:

Thank you very much for your comments concerning our manuscript entitled ‘3D Locating System for Pests’ Laser Control Based on Multi-constraint Stereo Matching’ (ID: agriculture-1700974). Those comments (at the bottom of the cover letter) are all valuable and very helpful for revising and improving our paper, as well as the important guiding significance to our researches. We have studied comments carefully and have made correction which we hope meet with approval.

Since the cover letter contains some figures, please see the attachment.

The above are my main corrections in the paper and the responds to the Reviewer’s comments. Special thanks to you for your comments. We tried our best to improve the manuscript and made some changes in the manuscript. Any revised portion made to the manuscript are marked up using the “Track Changes” function. At the same time, we appreciate for Editors and Reviewers warm work earnestly, and hope that the correction will meet with approval.

Once again, thank you very much for your comments and suggestion. 

Yours sincerely,

Yajun Li

Reviewer 3 Report

We congratulate the authors for their paper. The paper is well structured and we only recommend:

1) Mention the scientific name of the pest insect for the first time in the following form:

Pieris rapae (L.) (Lepidoptera: Pieridae)

Subsequently, it can be cited in summary form:

P. rapae

2) Restructure figures 5, 11 and 19 in order to improve the visualization of the tags.

Author Response

Dear Reviewer:

Thank you very much for your comments concerning our manuscript entitled ‘3D Locating System for Pests’ Laser Control Based on Multi-constraint Stereo Matching’ (ID: agriculture-1700974). Those comments (at the bottom of the cover letter) are all valuable and very helpful for revising and improving our paper, as well as the important guiding significance to our researches. We have studied comments carefully and have made correction which we hope meet with approval.

Since the cover letter contains some figures, please check the attachment.

The above are my main corrections in the paper and the responds to the Reviewer’s comments. Special thanks to you for your comments. We tried our best to improve the manuscript and made some changes in the manuscript. Any revised portion made to the manuscript are marked up using the “Track Changes” function. At the same time, we appreciate for Editors and Reviewers warm work earnestly, and hope that the correction will meet with approval.

Once again, thank you very much for your comments and suggestion. 

Yours sincerely,

Yajun Li
